# Relationship between Vitamin D3 Deficiency, Metabolic Syndrome and *VDR, GC*, and *CYP2R1* Gene Polymorphisms

**DOI:** 10.3390/nu16091272

**Published:** 2024-04-25

**Authors:** Carmina Mariana Stroia, Timea Claudia Ghitea, Maria Vrânceanu, Mariana Mureșan, Erika Bimbo-Szuhai, Csaba Robert Pallag, Annamaria Pallag

**Affiliations:** 1Doctoral School of Biomedical Sciences, Faculty of Medicine and Pharmacy, University of Oradea, 410073 Oradea, Romania; stroia.carminamariana@student.uoradea.ro; 2Pharmacy Department, Faculty of Medicine and Pharmacy, University of Oradea, 410073 Oradea, Romania; timea.ghitea@csud.uoradea.ro; 3Department of Toxicology, Iuliu Hatieganu University of Medicine and Pharmacy Cluj Napoca, 400012 Cluj-Napoca, Romania; marievranceanu@gmail.com; 4Department of Preclinical Disciplines, Faculty of Medicine and Pharmacy, University of Oradea, 410073 Oradea, Romania; mmuresan@uoradea.ro; 5Department of Surgery, Faculty of Medicine and Pharmacy, University of Oradea, 410073 Oradea, Romania; bszera@gmail.com; 6MSc International Economy and Business Program of Study, Department of World Economy, Corvinus University of Budapest, 1093 Budapest, Hungary; csaba.pallag@stud.uni-corvinus.hu

**Keywords:** vitamin D3 supplementation, metabolic syndrome, BMI, overweight, obese, 25(OH)D3, *CYP2R1*, *VDR*, *GC* gene polymorphism

## Abstract

The presence of vitamin D3 deficiency associated with the presence of metabolic syndrome (MS) has important public health effects. This study aims to investigate the relationship between vitamin D3 deficiency, MS and vitamin D3 receptor (*VDR*), GC Vitamin D binding protein (*GC*), and cytochrome P450 family 2 subfamily R member 1 (*CYP2R1*) gene polymorphisms, and genes whose encoded proteins are responsible for vitamin D3 metabolism and transport. A total of 58 participants were included in this study (age 39 ± 12 years) and were selected over a 12-month period. They were divided into four groups, depending on the presence of polymorphisms in *VDR*, *GC*, and *CYP2R1* genes and their weight status. At baseline, in months 3, 6, and 12, biochemical parameters including 25(OH)D3, total cholesterol, LDL cholesterol, HDL cholesterol, triglycerides, and homeostatic model assessment (HOMA index), the insulin resistance indicator were measured. Our results show that all subjects in the polymorphism group supplemented with vitamin D3 reached an optimal level of vitamin D3 associated with high concentrations of 25(OH)D3. Weight loss was most significant in patients in the POW group (overweight patients).

## 1. Introduction

Vitamin D3 deficiency is a global public health issue. Vitamin D3 is a fat-soluble vitamin and a steroid hormone involved in calcium and phosphorus homeostasis [1]. Vitamin D3 (cholecalciferol) is the natural form of vitamin D3, and the body can synthesize it in the skin in response to sunlight exposure. Diet is another important source of vitamin D3—fish oil, egg yolks, liver, dairy products, and for vegans, sources of ergocalciferol (D2) include mushrooms and yeast. Vitamin D3 deficiency is defined as a serum concentration of 25-hydroxyvitamin D3 (25(OH)D3) below 50 nmol/L or 20 ng/mL and has been associated with numerous disorders such as cardiovascular diseases, hypertension, dyslipidemia, type 2 diabetes, cancer, and others [2].

Interest in this vitamin has intensified in recent years due to the high prevalence of vitamin D3 deficiency, which is described as a worldwide epidemic. Based on vitamin D3 levels, there is a classification of insufficiency when 25(OH)D3 has plasma levels between 21 and 29 ng/mL, mild deficiency when levels are between 10 and 20 ng/mL, moderate deficiency between 9 and 5 ng/mL, and severe deficiency when vitamin D3 levels are less than 5 ng/mL.

Obesity is a multifactorial metabolic disorder whose prevalence has been proven to increase at an alarming rate. It is defined as an excess amount of body fat and represents a significant global health issue. The association between vitamin D3 deficiency and obesity, as well as with obesity-related diseases, has been confirmed by numerous studies, but the presence of a causal relationship is still unclear. There are many possible explanations regarding the inverse relationship between increasing adiposity, especially abdominal obesity, and low plasma concentrations of vitamin D3, but so far, none of these hypotheses have been able to elucidate this relationship fully. Therefore, multiple mechanisms may impact the interaction between vitamin D3, obesity, and associated diseases. Several genes have been implicated in the vitamin D3 metabolic pathway, and the genetic variants of these genes are associated with obesity-related phenotypes and vitamin D3 deficiency [3,4,5]. Numerous clinical studies have shown that vitamin D3 supplementation reduces the level of metabolic parameters such as total cholesterol, low-density lipoproteins (LDL), and triglycerides and decreases the insulin resistance indicator, homeostatic model assessment (HOMA index)—in patients with type 2 diabetes. However, it is not fully understood how vitamin D3 can reduce the risk of metabolic disorders. Recently, the vitamin D3 receptor (VDR) and vitamin D3 metabolizing enzymes have been detected in various types of cells, including pancreatic β cells and insulin-responsive cells such as adipocytes. Adipose tissue is a major storage site for vitamin D3 and an important source of adipokines and cytokines involved in systemic inflammation [6], but also at the level of certain parts of the body, including the gingival area [7]. It has also been suggested that the potential link between diabetes and obesity is vitamin D3 deficiency coexisting with obesity. Studies have identified several genetic variants in the *VDR*, *GC*, and *CYP2R1* genes that are associated with circulating levels of 25-hydroxyvitamin D3 (25(OH)D3) and vitamin D3 deficiency in Western populations. Given the current increase in the incidence of overweight individuals in developed countries, obesity appears to be a noteworthy factor among the numerous risk factors associated with identified vitamin D3 deficiency [8,9].

The vitamin D3 receptor (*VDR*) gene is located on chromosome 12q, position 13, contains 11 exons, and spans approximately 75 kb. The *VDR* gene encodes the VDR protein, the receptor for vitamin D3 that binds to the active form of vitamin D3, known as calcitriol. Activation or deactivation of this gene controls the absorption of calcium and phosphate, as well as other metabolic processes. Activated vitamin D3 exerts its cellular functions after binding to the nuclear vitamin D3 receptor (*VDR*) [10]. The VDR gene harbors more than 900 allelic variants, a fraction of which are believed to interfere with Vitamin D3 function. In total, six functional SNPs from regulatory regions in the VDR gene were selected for the analysis, namely rs739837, rs4516035, rs11568820, rs731236, rs2228570, and rs1544410. All of these SNPs had a function already reported in the literature or predicted by the SNP function prediction tool [11]. For the VDR gene, the rs731236 variant (T > C) is commonly known as TaqI and is located in exon 9. The normal variant of the gene is the homozygous T/T genotype, the mutant heterozygous variant is T/C, and the homozygous variant is C/C, which is predisposed to lower levels of vitamin D3. Although the relationship between vitamin D3 insufficiency and components of metabolic syndrome has been previously demonstrated [2,12], few studies have examined *VDR* gene polymorphisms for associations with the risk of these disorders [13,14,15]. Additionally, 25(OH)D3 levels were significantly lower in the T/C genotype compared with the C/C genotype [16].

Cytochrome P450 Family 2 Subfamily R Member 1 (*CYP2R1*) gene is located on chromosome 11p15.2 (gene ID: 120227), contains five exons, with approximately 15.5 kb, and encodes 501 amino acids [17]. *CYP2R1* gene encodes vitamin D3 25-hydroxylase, which is considered the most important enzyme in vitamin D3 metabolism and is a member of the cytochrome P450 enzyme superfamily, frequently involved in the metabolism of foreign compounds [18]. The enzyme catalyzes the transformation of vitamin D3 into an active ligand for the vitamin D3 receptor. A mutation in this gene has been associated with a selective deficiency of 25 (OH)D3 [19]. Several *CYP2R1* gene polymorphisms are functional and affect gene expression. Four single nucleotide polymorphisms (SNPs) have been identified for the *CYP2R1* gene: rs1562902, rs7116978, rs10741657, and rs10766197, which may predispose to lower serum concentrations of 25 (OH)D3 [20]. Vitamin D3 25-hydroxylase is the key enzyme that activates vitamin D3, which is obtained through sunlight exposure and diet. For the *CYP2R1* gene, the analyzed variants in this study were rs10741657. The wild-type genotype of the gene is A/A, the mutant heterozygous genotype is G/A, and the mutant homozygous genotype is G/G. A meta-analysis including 16 articles with 52,417 participants showed that the *CYP2R1* gene SNP rs10741657, with G/G homozygous genotype, presents trends of lower serum calcydiol levels compared with the A/A homozygous genotype in Caucasian and Asian populations [13]. In apparently healthy subjects, with SNP rs10743657 genotypes containing the allele A (A/A, G/A genotypes) were associated with higher levels of 25 (OH)D3 than the homozygous genotype (G/G), concluding that the G/G genotype predisposes to the lowest levels of vitamin D3 [15].

The GC vitamin D binding protein gene *GC* gene is located on the long arm of chromosome 4 (4q12–q13) with a size of ~58 kD, contains 13 exons and 12 introns, and encodes the vitamin D3-binding protein. It contains 474 amino acids encoded by 1690 nucleotides [17]. The *GC* gene is a polymorphic gene that belongs to the albuminoidal superfamily along with albumin, alpha-fetoprotein, and afamin. More than 120 variants of the GC gene have been described, but there are only two single nucleotide polymorphisms (SNPs) that give rise to three common haplotypes with distorted geographical distributions that appear to correlate with skin pigmentation, sunlight exposure, and possibly with the binding of vitamin D3 metabolites [13]. The rs2282679 SNP of the *GC* gene has been associated in multiple studies with low serum concentrations of vitamin D3. Therefore, the potential for vitamin D3 insufficiency is linked to the presence of rs2282679 (C/C). Carriers of two such alleles (C/C) have lower levels of vitamin D3 than carriers of one allele (C/A) [21]. For the rs2282679 variant, the wild-type genotype is A/A, which presents normal levels of vitamin D3 in the blood. The mutant alleles are A/C and C/C, among which the A/C genotype is associated with average, slightly decreased levels of serum vitamin D3. The C/C genotype is associated with the lowest values of serum vitamin D3 levels. The function of the vitamin D3-binding protein is to transport vitamin D3 to tissues; therefore, polymorphisms in the encoding gene variant rs2282679 may be associated with an increased risk of low circulating levels of vitamin D3. GC polymorphisms have been associated with susceptibility to some diseases, such as diabetes and obesity [22].

Starting from the analysis of the presence of genetic polymorphisms at the level of the main genes whose encoded proteins are involved in the metabolism and transport of vitamin D3, we analyzed the effect of vitamin D3 supplementation in patients with vitamin D3 deficiency and the effect of the optimal level of vitamin D3 on metabolic syndrome.

Metabolic syndrome (MS) is defined by the presence of three of the following criteria (according to the consensus of major international institutions from 2009): the presence of abdominal obesity (abdominal circumference over 80 cm in women and over 94 cm in men); high triglyceride levels ≥ 150 mg/dL (1.7 mmol/L) or treatment for high triglyceride levels; low HDL-cholesterol levels <40 mg/dL (1 mmol/L) in men and <50 mg/dL (1.3 mmol/L) in women or treatment for low HDL: systolic blood pressure values ≥ 130 mmHg and/or diastolic blood pressure values ≥ 85 mmHg or treatment for hypertension; elevated fasting blood glucose levels ≥ 100 mg/dL or treatment for previously diagnosed diabetes mellitus [23,24,25].

## 2. Materials and Methods

This study was conducted between 30 July 2022 and 30 July 2023 after obtaining the approval of the Research Ethics Committee of the Faculty of Medicine and Pharmacy, University of Oradea, by Decision No. CEFMF/2 of 29 July 2022, along with the favourable opinion of the Ethics Committee of the “Dr. Ghe. Marinescu” Municipal Hospital Târnăveni No. 18766/15 July 2022, and of the Nutrition Cabinet, SC CSBNDiet Precision NUTRITION SRL No. 1/15 July 2022. All patient data were processed in accordance with the Code of Medical Deontology in Romania and in accordance with the Helsinki Agreement.

Recruitment of study participants was carried out at the nutrition cabinet, SC CSBNDiet Precision NUTRITION SRL; thus, 48 adults with polymorphisms in the *VDR*, *CYP2R1*, and *GC* genes, and 10 adults without polymorphisms in those genes were included in this study, as a control group.

### 2.1. Study Design

The inclusion and exclusion criteria were as follows: Inclusion criteria: age 18 or over, both females and males, Romanian citizens; Exclusion criteria: minors, pregnant and lactating women, patients with psychiatric problems, oncology subjects, patients with various chronic conditions.

Four study groups were organized: the control group (CG), including 10 subjects without *VDR*, *GC*, or *CYP2R1* gene polymorphisms who followed a genetic diet; the POW study group, including 16 subjects with *VDR*, *GC*, and *CYP2R1* gene polymorphisms overweight, who followed a genetic diet and vitamin D3 supplementation; the PO study group, including 16 subjects with *VDR*, *GC*, and *CYP2R1* gene polymorphisms, obese, who followed a genetic diet and vitamin D3 supplementation; the PWS study group, including 16 obese and overweight subjects with *VDR*, *GC*, and *CYP2R1* gene polymorphisms, who followed a genetic diet, without vitamin D3 supplementation. The genetic diet plan consists of either three main meals and two snacks or only three main meals, depending on the patient’s individual needs. Macronutrients and micronutrients are calculated to meet specific nutritional requirements, also taking into account genetic polymorphisms.

The demographic overview of this study’s groups includes average age, gender distribution, urban or rural origin, the evaluation of the presence of genetic polymorphisms in *VDR*, *GC*, and *CYP2R1* genes, initial 25(OH)D3 levels, and initial BMI values.

After entering this study, oral swab collection and saliva DNA sampling for genetic testing took place in a single visit to the Nutrition Cabinet. The kits were shipped to the genetic laboratory at Nkaarco Diagnostics Limited, Bakersfield, Norwich, UK. The first set of laboratory medical tests was conducted upon receipt of the genetic test results, followed by the repetition of monitored parameters at intervals of 3, 6, and 12 months. Patients from the POW and PO groups received a personalized scheme with Vitamin D3 supplementation, based on the 25(OH)D3 levels, as follows in Table 1.

For patients included in the POW and PO groups with a 25 OH Vitamin D3 value between 10 and 19.99 ng/mL, a total of 400,000 IU of vitamin D3 was administered, divided into 4 doses over 8 weeks, with maintenance every 3 months, with a 100,000 IU ampule. Patients with 25 OH vitamin D3 values between 20 and 49.99 ng/mL receive additional vitamin D3 supplementation with ampules totaling 300,000 IU, and upon normalization of the value, they receive the maintenance dose every 90 days of 100,000 IU.

### 2.2. Highlighting Gene Polymorphisms in VDR, CYP2R1, and GC Genes

Within this study, three SNPs related to vitamin D3 metabolism (*VDR*—rs731236 SNP, CYP2R1—rs10741657 SNP, and GC—rs2282679 SNP) were selected based on recent genome-wide association studies on 25-hydroxyvitamin D3. For testing DNA samples collected from this study’s participants, the IsohelixBuccalyse DNA extraction kit BEK-50 (Kent, UK) was used for DNA extraction, and PCR was performed on an ABI7900 real-time thermocycler (Applied Biosystem, Waltham, MA, USA) to determine the SNPs in the selected genes.

#### Description of the Working Method

Buccal cells were collected from the lateral wall of the oral vestibule using the commercial Buccal swab with Rapidri Pouch Isohelix kit, following the manufacturer’s recommendations. The samples were couriered to the genetic laboratory (Nkaarco Diagnostics Limited, Badersfield, Norwich, UK). Subsequently, DNA was extracted from the buccal cells collected on swabs—a standard method for BEK-3/BEK-50, followed by genotype analysis using the ABI7900 real-time thermocycler system, optimized for use with Applied Biosystems chemistry, including nucleic acid quantification and detection. In detail, 10–50 ng of genomic DNA was amplified in a total volume of 10 μL. TaqMan Genotyping Mastermix (Applied Biosystems, Waltham, MA, USA), 0.5 μL of 20× TaqMan respective genotyping assay, and 10 μL purified water were used. The PCR conditions were as follows: 10 min at 95 °C (initial denaturation), followed by 40 cycles alternating at 95 °C for 15 s (denaturation) and 60 °C for 60 s (annealing/extension) on a rapid real-time thermocycler Applied Biosystems 7900HT (Applied Biosystems, Waltham, MA, USA). The genotypes were analyzed using the Sequence Detection Systems (SDS) software, version 2.2.1 (Applied Biosystems, Waltham, MA, USA). The following SNPs were analyzed: *VDR*—rs731236 (A > G), *CYP2R1*—rs10741657 (A > G), and *GC*—rs2282679 (A > C) [26].

### 2.3. Biochemical Markers

The analyzed biochemical markers were 25(OH)D3 level, total serum cholesterol, LDL cholesterol, HDL cholesterol, triglycerides, and HOMA index. After participants had fasted for 12 h, blood samples were obtained from an antecubital vein while participants were sitting. A tourniquet was used, but it was released before the withdrawal of blood into Vacutainer tubes (Becton Dickinson, Mountain View, CA, USA) containing EDTA. Plasma was separated from blood cells by centrifugation and immediately used for the measurement of biochemical markers. The collected blood samples were venous blood. The collection container was vacutainer without anticoagulant and gel separator. After collection, the serum is separated by centrifugation; it is processed on the same day with the Konelab 60i automated analyzer (Thermo Scientific, Vantaa, Finland). In the Tosoh Bioscience automatic analyzer (Tosoh Corporation Bioscience Division, Tokyo, Japan) and the Atellica IM analyzer (Siemens Healtcare Diagnostic INC Laboratory Diagnostic, Tarrytown, NY, USA), the serum has a stability of 4 days at 2–8 °C, and 1 month at −20 °C.

#### 2.3.1. Determination of the 25(OH)D3 Level

The determination of the 25(OH)D3 level is the best indicator of vitamin D3 status in the body, with a half-life of 2–3 weeks. Over 90% of vitamin D3 is represented by 25(OH)D3. Vitamins D3 or calciferol are synthesized from provitamins by cleaving the B ring of the sterol molecule during exposure to sunlight. The most important vitamin D forms are vitamin D2 (ergocalciferol) and vitamin D3 (cholecalciferol). Vitamin D2 is the plant-derived form (ergosterol or provitamin D2). Vitamin D3 comes either from animal-derived foods (especially fatty fish or fish oil) or nutritional supplements, or it is synthesized in the skin from 7-dehydrocholesterol (provitamin D3) under the action of ultraviolet radiation. The rate of vitamin D3 formation depends mainly on the duration and intensity of exposure. The ST AIA-PACK 25-OH Vitamin D3 reagent (Tosoh Bioscience, Inc., San Francisco, CA, USA) and spectrophotometer (TOSOH 900, Tosoh Bioscience, In., San Francisco, CA, USA) were used to measure the 25(OH)D3 level, using reference values: ≤20 ng/mL, deficiency; 21–29 ng/mL, insufficient level; 30–100 ng/mL optimal level of 25(OH)D3.

#### 2.3.2. Determination of Total Serum Cholesterol Level

Determining the level of serum cholesterol assesses lipid status and metabolic disorders, the risk of atherosclerosis, coronary stenosis, and myocardial infarction. Cholesterol is a fat-like substance (lipid) that is present in cell membranes and is a precursor of bile acids and steroid hormones. Cholesterol travels in the blood in distinct particles containing both lipids and proteins (lipoproteins). Three major classes of lipoproteins are found in the serum of a fasting individual: low-density lipoproteins (LDL), high-density lipoproteins (HDL), and very low-density lipoproteins (VLDL). To measure the total serum cholesterol, the used reagents are produced by Diagnosticum Zrt, Budapest, Hungary; biochemistry analyzer Konelab 60i (Thermo Scientific, Vantaa, Finland). The spectrophotometric method (enzymatic–colorimetric) was used. Reference values for total serum cholesterol are <200 mg/dL optimal level; 200–240 mg/dL borderline level; >240 mg/dL high level [27].

#### 2.3.3. Determination of HDL Cholesterol Level

HDL plays an important role in cholesterol metabolism, participating in transporting it from extrahepatic tissues to the liver for catabolism and excretion [28,29]. HDL cholesterol concentrations and apolipoprotein-A levels are positive risk factors for atherosclerosis. Patients with normal and high levels of HDL cholesterol in the blood are protected, with a low risk of atherosclerosis and cardiovascular diseases [30].

HDL Cholesterol–R1, R2 enzymatic—HDL Cholesterol reagents, produced by Diagnosticum Zrt, Budapest, Hungary and biochemistry analyzer Konelab 60i (Thermo Scientific, Vantaa, Finland) was used for spectrophotometric method (enzymatic–colorimetric). Reference values for HDL cholesterol level are <40 mg/dL normal value; >60 mg/dL, protective value; <40 mg/dL. Clinical alert value, which is associated with an increased risk of coronary artery disease.

#### 2.3.4. Determination of LDL Cholesterol Level

LDL is the lipoprotein that contains the highest amount of cholesterol (60–70% of total serum cholesterol). LDL cholesterol’s role is involved in transporting cholesterol to tissues, mainly in the arterial system, which explains the increased incidence of atherosclerosis and coronary artery disease in patients with high serum levels of this lipoprotein. Thus, determining LDL cholesterol levels is specific for estimating cardiovascular risk and making therapeutic decisions. LDL cholesterol contains Apo B-100 as the associated apolipoprotein (20% of the total content), with a density of 1.019 to 1.063 g/mL and a diameter of 20 to 25 nm. Small LDL particles are more atherogenic than larger ones [31].

In addition, dietary cholesterol intake is usually associated with increased intake of saturated fatty acids, which is documented to increase LDL cholesterol and the risk of cardiovascular disease [32]. R1, R2 enzymatic—LDL Cholesterol reagents, produced by Diagnosticum Zrt, Budapest, Hungary, and a spectrophotometric method, analyzer Konelab 60i (Thermo Scientific, Vantaa, Finland) were used.

Reference values for LDL cholesterol: <100 mg/dL, optimal level; 100–129 mg/dL, near-optimal level; 130–159 mg/dL, borderline high level; 160–189 mg/dL, high level; ≥190 mg/dL, very high level.

#### 2.3.5. Determination of Triglyceride Levels

Triglycerides in adipose tissue and other tissues represent the most important energy reserve deposits in the body. In adipose tissue, they are stored as glycerol, fatty acids, and monoglycerides, which are converted into triglycerides in the liver, entering the composition of VLDL (80%) and LDL (15%). Hypertriglyceridemia, together with hypercholesterolemia, are independent risk factors for atherosclerotic disease. For the determination of triglyceride levels, the R1 enzymatic—triglyceride reagent, produced by Diagnosticum Zrt, Budapest, Hungary, and a spectrophotometric method was used, Konelab 60i (Thermo Scientific, Vantaa, Finland). Reference values for serum triglycerides: <150 mg/dL, optimal level; 150–199 mg/dL, near-optimal level; 200–499 mg/dL, borderline high level; ≥500 mg/dL, very high level.

#### 2.3.6. Determination of HOMA Index

The homeostatic model assessment (HOMA index), the insulin resistance indicator, is calculated from basal levels of glucose and insulin according to the formula: HOMA-IR = (insulin (µU/mL) × glucose (mg/dL))/405. The serum insulin was determined by the immunochemical method with photometry detection of insulin (Atellica IM Insulin Reactive reagent, Siemens, Atellica, Germany). Blood glucose levels were determined using the spectroscopic method (enzymatic colorimetric) and glucose PAP reagent (Diagnosticum Zrt, Budapest, Hungary). Reference values for HOMA index: <2: normal; >2: possible insulin resistance; >2.5: an increased probability of insulin resistance; >5: average value in diabetic patients [33,34,35,36].

### 2.4. Anthropometric Measurements

Anthropometric parameters monitored throughout this study included height, weight, abdominal circumference, and body mass index (BMI = weight (kg)/height (m^2^)) [37].

Monitoring of laboratory biochemical parameters and anthropometric measurements was conducted at the beginning of this study, at 3 months, 6 months, and 12 months from the start of the genetic diet and supplementation with vitamin D3, according to Table 2.

### 2.5. Statistical Analysis

Analysis of these data was conducted using the statistical software SPSS 20 (New York, NY, USA), employing various statistical techniques, including analysis of variance (ANOVA), post hoc analysis, the Chi-square test, and inferential statistics such as the Student’s *t*-test. Comparison among the three research groups was performed using the Bonferroni test. Correlations between parameters were assessed utilizing Bravais–Pearson tests and paired sample correlation.

## 3. Results

### 3.1. Baseline Characteristics

This study involved a total of 58 patients, including 12 males (20.7%) and 46 females (79.3%), with an average age of 41 years (41 ± 12). Among the subjects, 23 were obese (BMI over 30 kg/m^2^) 2 (54.1 ± 6.6), 25 were overweight (BMI over 25 kg/m^2^) (54.1 ± 6.6), and the remaining 10 (54.1 ± 6.6) were normal weight (Table 2).

### 3.2. Genotype Distributions and Allele Frequencies

Genotype distributions and allele frequencies of the studied VDR, CYP2R1, and GC gene SNPs are shown in Table 3. The VDR rs731236 T > C SNP were studied. The homozygous T/T was considered the reference genotype, which predisposes to normal levels of Vitamin D3, heterozygous T/C, and homozygous C/C genotypes which predispose to lower levels of vitamin D3. At the same time, the CYP2R1 rs10741657 A > G SNP was studied, where A/A homozygous and A/G heterozygous genotypes predispose to normal levels of Vitamin D3, homozygous G/G genotypes predispose to lower levels of vitamin D3. In the case of the GC gene, the rs2282679 A > C SNP was studied. The A/A homozygous genotype was considered the reference genotype. A/C heterozygous and C/C homozygous genotypes predispose to lower levels of vitamin D3.

In Table 4, Table 5 and Table 6, data on the frequency of each polymorphism out of the total of 58 subjects are presented. Each genotype is listed together with its allele frequency in this study’s groups.

As shown in Table 4 the analysis of polymorphism frequency provides important information about the distribution of different genotypes of the *VDR* gene in the studied groups. The C/T heterozygous and C/C homozygous genotypes (62.1%) dominate in terms of their frequency among the studied subjects, at the detriment of the frequency of the T/T wild type genotype (37.9%).

The analysis of genotype distribution on the *CYP2R1* gene SNP provides essential information about the genetic diversity in the studied groups. It is evident that the A/G heterozygous and G/G homozygous genotypes (63.8%) dominate in terms of their frequency among the studied subjects.

Table 6 provides detailed information about the *GC* gene SNP. The frequency of the A/A homozygous wild-type allele (58.6%) is higher among the analyzed subjects, followed by the A/C heterozygous genotype, while the C/C genotype is completely absent in this studied population.

These findings are essential for understanding the genetic diversity of the population and can provide important clues in research on genotype–phenotype associations and potential correlations with various health conditions or phenotypic characteristics. Genotype distribution of the studied genes SNPs shows that in the case of *VDR* and *CYP2R1* genes SNPs, the genotypes that predispose to lower levels of vitamin D3 are significantly frequent. Only in the case of the *GC* gene SNP is the homozygous A/A reference genotype more frequent than the A/C heterozygous genotype, which predisposes to lower levels of vitamin D3.

### 3.3. Body Mass Index (BMI)

Anthropometric measurements, particularly body mass index (BMI), serve as crucial indicators of weight management and metabolic health. In the context of diverse intervention strategies, understanding the trends in BMI across different study groups sheds light on the effectiveness of various interventions, including genetic dietary interventions and Vitamin D3 supplementation. This introduction explores the anthropometric measurements for BMI across four distinct study groups: GC, POW, PO, and PWS, each representing different intervention approaches and their potential impact on weight management and metabolic syndrome. Specifically, the GC group did not receive any dietary or supplementation intervention, while the POW and PO groups were subjected to genetic diet interventions and Vitamin D3 supplementation. In contrast, the PWS group solely underwent genetic diet intervention. Analyzing BMI trends within these groups offers valuable insights into the efficacy of intervention strategies and their association with metabolic syndrome risk factors. In group GC, the initial BMI of 22.83 remained mostly unchanged at 12 months, with a value of 22.90, indicating no significant changes in weight within this group over the study period. This suggests that individuals in group GC maintained a relatively stable weight throughout this study. In group POW: Participants in this group started with a higher initial BMI ratio of 27.14. After supplementation with Vitamin D3 for 12 months, the group’s BMI decreased by 2.96 units to 24.18. In the group, PO has demonstrated the most significant improvement in BMI. Starting with a high initial BMI of 34.99, participants achieved a substantial decrease to 28.92 at 12 months. In group PWS: Participants in this group began with a BMI of 29.58, which decreased to 26.41 after one year.

Table 7 shows the evolution of BMI over the studied 12 months in the studied groups. The table displays the mean BMI values at four time points: initial, 3 months, 6 months, and 12 months.

The therapeutic intervention applied led to weight loss in all groups where it was applied, with those in the GC group showing no significant changes in BMI. Particularly, the groups that benefited from supplementation with vitamin D3 recorded the greatest effect in reducing body mass index (BMI) and abdominal circumference.

Patients from the POW group had the highest rate of weight normalization, with 98% of them becoming normal weight.

In the PO group, only 5% of patients achieved normal weight, while the rest experienced weight loss, falling into the overweight or obesity category with a lower BMI. In the PWS group, which combines characteristics of both obesity and overweight, 45% of patients reached normal weight, while the rest were categorized as overweight or obese with a reduced BMI.

These results suggest that vitamin D3 supplementation may play a significant role in weight reduction and improving body composition, with positive implications for metabolic health and the risk of metabolic syndrome. It is important to emphasize that these findings demonstrate the effectiveness of a personalized approach in weight management and metabolic risk.

### 3.4. Abdominal Circumference

Significant variations in abdominal circumference were observed among the different study groups. In the GC group, the mean abdominal circumference remained relatively stable, with a slight decrease from 74.4 cm at the beginning of this study to 74.2 cm at the end of this study. In the POW group, a significant decrease in abdominal circumference was recorded, from an initial average of 86.43 cm to 79.62 cm at the end of the 12-month study period, indicating the positive effect of the interventions applied. The PO group also showed a significant reduction in abdominal circumference, from 94.28 cm at the beginning of this study to 84.03 cm at the end, suggesting the effectiveness of dietary and supplementation interventions. In the PWS group, a significant decrease in abdominal circumference was recorded, from an initial average of 92.18 cm to 84.5 cm at the end, indicating the benefits of genetic dietary intervention.

These findings underscore the importance of personalized interventions in weight management and suggest that both genetic diet and vitamin D3 supplementation can contribute to reducing abdominal circumference and, implicitly, metabolic risk.

Table 8 presents the mean values of abdominal circumference at four time points: initial, 3 months, 6 months, and 12 months for each group.

### 3.5. Determination of 25(OH)D3 Level

The assessment of the 25(OH)D3 level in the studied groups (GC, POW, PO, and PWS) highlights different trends depending on the interventions, as shown in Figure 1A: In the GC group the initial average value for the 25(OH)D3 level was 38.84 ng/mL, and at 12 months, a decrease was recorded to an average value of 36.6 ng/mL of the 25(OH)D3 level. In the POW group, participants started with an initial average value of the 25(OH)D3 level of 17.58 ng/mL, and after 12 months of supplementation, a significant improvement was recorded, with an average value of 67.19 ng/mL. In the PO group, the initial values were close to those of the POW group, showing the link between vitamin D3 and vitamin D3 deficiency, with an average of 17.65 ng/mL of the 25(OH)D3 level, but at 12 months, a significant increase was recorded, with an average of 61.72 ng/mL. In the PWS group, the initial value of the 25(OH)D3 level was lower, 15.5 ng/mL, and at 12 months, a slight decrease was recorded, with an average value of 14.74 ng/mL of the 25(OH)D3 level.

### 3.6. Determination of Total Serum Cholesterol

The evolution of the biochemical parameter of total serum cholesterol within the four studied groups (GC, POW, PO, and PWS) provides an insight into how different interventions through supplementation with vitamin D3 and genetic diet have impacted it (Figure 1B).

In the GC group, although no major changes were recorded, the average value of total serum cholesterol showed a slight decrease from 186.4 at the beginning of this study to 184.1 mg/dL at the end of it. In the POW group, a significant reduction in total serum cholesterol was observed, from a high initial average value of 217.87 mg/dL at the beginning of this study to 165.81 mg/dL at 12 months, indicating the effectiveness of the intervention through genetic diet and supplementation with vitamin D3 in managing this parameter. The PO group also showed a significant decrease in total serum cholesterol, from an initial average of 213.56 mg/dL to a final average of 160.56 mg/dL, suggesting the positive impact of the applied interventions on cardiovascular health. In the PWS group, a significant reduction in total serum cholesterol was recorded, from a high initial average of 254.18 mg/dL to a final average of 213.43 mg/dL at 12 months, suggesting that genetic diet had a positive impact on the lipid profile of the subjects.

### 3.7. Determination of HDL Cholesterol

The evolution of the HDL cholesterol within the studied groups (GC, POW, PO, and PWS) can provide insight into the effects of intervention through supplementation with vitamin D3 and genetic diet, as shown in Figure 1C. In the GC group, the average HDL cholesterol value remained relatively stable, with a slight decrease from 54.92 mg/dL at the study entry to 54.28 mg/dL at its conclusion. In the POW group, a significant increase in HDL cholesterol was recorded, from an initial average value of 47.83 mg/dL at the beginning of this study to 56.88 mg/dL at 12 months, indicating the positive impact of the intervention through genetic diet and supplementation with vitamin D3 on this indicator of cardiovascular health. Regarding the PO group, minimal variation in the level of HDL cholesterol was observed, with a slight increase from 51.94 mg/dL at the first laboratory analysis to 52.14 mg/dL at the end of this study. In the PWS group, there was also a significant increase in HDL cholesterol, from an initial average of 48.18 mg/dL to an average of 55.02 mg/dL at 12 months, suggesting the benefits of intervention through a genetic diet.

### 3.8. Determination of LDL Cholesterol

The evolution of the LDL cholesterol level in the studied groups (Figure 1D) provides insight into how different interventions or conditions can influence this marker of cardiovascular health. In the GC group, the average LDL cholesterol level was 112.79 mg/dL at the beginning of this study and 108.4 mg/dL at the end of it. In the POW group, a significant difference between the initial values and those at 12 months can be seen, with an initial average value of 129.61 mg/dL and a mean of 125.23 mg/dL at the end of this study. Regarding the PO group, we note a significant decrease in LDL cholesterol, from an initial average of 147.13 mg/dL to a final average of 118.62 mg/dL. In the PWS group, a significant decrease in LDL cholesterol level can be seen, with an initial average value of 153.61 mg/dL and a final average of 132.55 mg/dL at 12 months.

### 3.9. Triglycerides

The evolution of the triglyceride level in the four studied groups provides insight into how different interventions may influence this marker of metabolic health (Figure 1E). In the GC group, the average triglyceride value showed a slight decrease from 121.1 mg/dL at the beginning of this study to 109.4 mg/dL at its conclusion, indicating a potential improvement in lipid profile within this population during the study period. In the POW group, a significant decrease in triglycerides was observed, from an average value of 173.5 mg/dL at the beginning of this study to 116.81 mg/dL at 12 months, suggesting a positive impact of the interventions applied (genetic diet and vitamin D3 supplementation) on the triglyceride level in this group. Regarding the PO group, a decrease in triglycerides was also recorded, from an initial average of 183.18 mg/dL to a final average of 123.25 mg/dL, highlighting the effectiveness of the specific interventions applied in this study in reducing this marker of metabolic health. In the PWS group, a significant decrease in triglyceride levels was evident, from an initial average of 187.12 mg/dL to a final average of 142.69 mg/dL at 12 months, indicating the effectiveness of the genetic diet interventions applied in managing this aspect of lipid health.

### 3.10. HOMA Index

The evolution of the HOMA index within the studied groups provides insight into how different interventions or conditions can influence insulin sensitivity and insulin resistance in a given population (Figure 1F). In the GC group, the average HOMA index value showed a slight increase from 1.00 at the beginning of this study to 1.04 at its conclusion, suggesting a potential alteration in insulin sensitivity within this population during the study period. In the POW group, a significant difference was observed between the initial values and those at 12 months, with a substantial reduction in the HOMA index from an initial average value of 3.1 to 0.93 at the end of this study, indicating a positive impact of the interventions applied (genetic diet and vitamin D3 supplementation) on insulin sensitivity in this group. Regarding the PO group, a significant decrease in the HOMA index was recorded, from an initial average of 4.36 to a final average of 0.94, highlighting the effectiveness of the interventions applied in improving insulin sensitivity and managing metabolic disorders within this study. In the PWS group, a significant change in the HOMA index was also observed, with an initial average value of 3.63 and a final average of 1.7 at 12 months, indicating a positive impact of the specific interventions applied (genetic diet) on insulin sensitivity among participants.

### 3.11. Glycemia

The evolution of the glycemia level within the studied groups provides insight into how genotype-based dietary interventions and supplementation with vitamin D3 can influence blood sugar levels in a given population (Figure 2). In the GC group, the average blood glucose level showed a slight increase from 92.2 mg/dL at the beginning of the study to 95.6 mg/dL at its conclusion, suggesting a possible change in blood sugar levels in this population during the study period, but it remains within normal limits overall. In the POW group, a significant difference was observed between the initial values and those at 12 months, with a drastic reduction in blood glucose levels from an initial average of 121.5 mg/dL to 95.06 mg/dL at the end of this study, indicating a positive impact of the interventions applied (genotype-based diet and supplementation with vitamin D3) on blood sugar levels in this group. Regarding the PO group, a significant decrease in blood glucose levels was noted, from an initial average of 124.31 mg/dL to a final average of 94.93 mg/dL, highlighting the effectiveness of the interventions applied in managing blood sugar levels and preventing associated metabolic disorders. In the PWS group, a significant decrease in blood glucose levels was also observed, with an initial average of 121.43 mg/dL and a final average of 106.37 mg/dL at 12 months, indicating a positive impact of the specific interventions applied (genotype-based diet) on blood sugar levels among participants.

### 3.12. Insulin

The evolution of the insulin parameter within the studied groups (GC, CS, CO, and PWS) provides insight into how nutrigenetic interventions and supplementation with vitamin D3 can influence insulin levels in a given population (Figure 3). In the GC group, the mean insulin value was 4.49 mIU/L at the beginning of this study and 4.28 mIU/L at its conclusion. In the POW group, we observe a significant difference between the initial values and those at 12 months, with an initial mean value of 11.8 mIU/L and a mean value of 4 mIU/L at the end of this study. Regarding the PO group, we note a significant decrease in insulin levels, from an initial mean of 12.12 mIU/L to a final mean of 6.51 mIU/L. In the PWS group, we observe a significant decrease in insulin levels, with initial and final mean values of 12.12 mIU/L and 4.28 mIU/L, respectively, at 12 months.

### 3.13. The Pearson Correlation

In Table 9 it is shown the Pearson correlation. In the case of the *VDR* gene, the presence of rs731236 SNP with C/T and C/C genotypes causes low 25(OH)D3 levels, especially in the PO and POW groups (Figure 4A). In the case of the *CYP2R1* gene, the presence of rs10741657 SNP with G/G genotype causes low 25(OH)D3 levels, especially in the POW group (Figure 4B). In the case of the *GC* gene, the presence of rs2282679 SNP with A/C genotype causes low 25(OH)D3 levels, especially in the PO and POW groups (Figure 4C).

The Pearson correlation shows that the polymorphism at the *VDR* gene SNP level causes the most significant changes in 25(OH)D3 metabolism compared with the other two studied genes, *CYP2R1* and *GC*. These changes also affect the level of cholesterol, triglycerides, and glycemia.

## 4. Discussion

Throughout the 12-month study, encompassing four distinct study groups comprising a combined total of 58 subjects, notable reductions in weight were observed, particularly evident in the PO, POW, and PWS cohorts. While the GC and PWS groups adhered to a genetic diet regimen, the CO and POW groups received additional vitamin D3 supplementation by a predefined protocol alongside the genetic dietary intervention. The results indicate that with the implementation of a genetic diet and vitamin D3 supplementation according to the protocol, the PO and POW groups demonstrated the best response in terms of biochemical and anthropometric parameters compared with the PWS group, which only followed the genetic diet. These findings suggest that vitamin D3 supplementation may be beneficial in regulating the level of 25(OH)D3 and improving the lipid profile, including normalizing total cholesterol and LDL fraction, improving HDL levels, and decreasing triglycerides. Additionally, the genetic diet and vitamin D3 supplementation had a positive impact on anthropometric indices, including reducing BMI, which may contribute to reducing the risk of metabolic syndrome. Furthermore, an improvement in insulin sensitivity was observed, reflected by a decrease in the HOMA index, as well as a decrease in glucose and insulin levels. These findings suggest that for subjects with polymorphisms in genes involved in vitamin D3 metabolism and components of metabolic syndrome, adopting a personalized genetic diet and vitamin D3 supplementation may be effective strategies for achieving optimal metabolic health and reducing the risk of metabolic syndrome. These findings underscore the importance of personalized approaches in managing health, considering individual genetic variations and their interactions with environmental factors. For the studied genes *VDR*, *CYP2R1*, and *GC*, the level at which SNPs can influence 25(OH)D3 levels, it is noteworthy that the T/C variant for rs731236 (VDR), the G/G homozygous variant for rs10741657 (CYP2R1), and the homozygous variant for rs1800012 (C/C) are associated with lower vitamin D3 levels. In summary, while the GC group maintained a stable BMI, the POW, PO, and PWS groups experienced reductions in BMI over the 12-month study period. These findings suggest that interventions such as Vitamin D3 supplementation or specific study protocols may be effective in promoting weight loss or improving weight management in individuals with higher initial BMIs [38,39,40].

The results indicate that interventions such as genetic diet and vitamin D3 supplementation have shown significant effectiveness in reducing abdominal circumference and managing the risks associated with abdominal obesity and metabolic syndrome across different study groups. These findings underscore the potential benefits of such interventions in the management and prevention of abdominal obesity, highlighting the importance of personalized dietary approaches in promoting metabolic health.

In summary, notable enhancements in vitamin D3 levels were observed following supplementation in the POW and PO groups, whereas the GC and PWS groups demonstrated declines or lack of progress in this aspect. These results underscore the need for tailored supplementation strategies or interventions to address vitamin D3 deficiencies across various populations.

It can be observed that all groups experienced decreases in total serum cholesterol during this study, with the most significant reductions observed in the POW, PO, and PWS groups. These results indicate that interventions such as vitamin D3 supplementation and lifestyle changes—adopting the genetic diet applied in this study—had a positive effect on the level of total serum cholesterol among the participants.

By analyzing the evolution of HDL cholesterol levels in these studies, we can observe significant variations influenced by the applied interventions, such as vitamin D3 supplementation and adopting a personalized genetic diet, as well as the specific characteristics of the studied populations. Vitamin D3 supplementation and adopting a personalized genetic diet had a significant impact on HDL cholesterol levels in some study groups. However, for example, in the PWS group, a significant increase in HDL cholesterol levels was observed, possibly due to the lack of polymorphisms on the genes involved in HDL cholesterol metabolism and the implementation of the genetic diet. In contrast, in other groups, such as GC and PO, variations in HDL cholesterol levels were smaller or absent, suggesting that other genetic factors or interventions may be more relevant in these cases.

The genetic diet and vitamin D3 supplementation may have synergistic effects in reducing LDL cholesterol levels and improving the lipid profile. Observations from the studies support this hypothesis, highlighting that the combination of the genetic diet and vitamin D3 supplementation can be effective in lowering LDL cholesterol levels and improving cardiovascular health.

These combined interventions have the potential to influence multiple aspects of lipid metabolism and cardiovascular health, having a stronger effect than individual interventions. The genetic diet can be tailored to the individual’s needs and genetic characteristics, thus optimizing the response to food and lipid metabolism. Vitamin D3 supplementation can complement these effects, having a positive impact on LDL cholesterol levels and other lipid parameters.

Triglycerides are an important marker of metabolic health, and high levels are associated with an increased risk of metabolic syndrome and cardiovascular diseases. Therefore, reducing them can significantly contribute to reducing this risk and improving overall cardiovascular health.

The link between low triglyceride levels and metabolic syndrome is well documented in the literature. The reduction in triglycerides may be associated with a decrease in the risk of abdominal obesity, insulin resistance, and other components of metabolic syndrome.

This significant improvement may be attributed to the specific genetic intervention applied in this study, which had a positive impact on triglyceride levels.

In summary, by analyzing the evolution of the HOMA index in this study, we observe noteworthy variations depending on the interventions applied or the characteristics of the studied populations. These findings demonstrate the importance of the impact that various interventions, such as the genetic diet and vitamin D3 supplementation, can have on insulin sensitivity and, implicitly, on overall metabolic health. The results suggest that personalized interventions can play a crucial role in managing and preventing diabetes and other metabolic disorders. Understanding the factors that influence insulin sensitivity can contribute to the development of more effective intervention and treatment strategies. Thus, the results of this study could be relevant for the development of future therapeutic and preventive approaches in the field of diabetes and metabolic disorders, aiming to improve the metabolic health of the population and reduce the incidence of these conditions. Further research and investigations in this field are important to validate and consolidate the obtained results and to develop more precise and efficient strategies for the prevention and management of these metabolic diseases.

By analyzing the evolution of blood glucose in these studies, we observe significant variations depending on the interventions applied. These findings support the importance of personalized interventions in glycemic management and metabolic health.

Regarding metabolic syndrome (MS), with the various complications of MS, the results indicate that specific interventions, such as a genetic diet and/or supplementation with vitamin D3, can have a significant impact on parameters associated with MS, such as blood glucose levels, cholesterol, triglycerides, and insulin sensitivity. These findings suggest that personalized approaches, considering individual genetic and nutritional characteristics, can be crucial in managing and preventing MS.

After analyzing the evolution of insulin levels in this study, it becomes evident that both the interventions applied and the individual characteristics of the studied populations exerted a significant influence on insulin sensitivity and glucose metabolism. Through personalized interventions, such as a genetic diet and supplementation with vitamin D3, significant changes in insulin levels have been observed in different study groups. This suggests that personalized approaches are essential in managing metabolic health and the risk of diabetes and other associated metabolic disorders. It is important to note that further studies could investigate in more depth the mechanisms involved and evaluate the long-term impact of these personalized interventions on metabolic health and the risk of metabolic diseases. These findings can serve as a basis for developing efficient prevention and treatment strategies in the field of diabetes and other associated metabolic disorders.

There have been several studies regarding the effects of vitamin D3 status and vitamin D3 supplementation on weight loss and associated parameters over time, Zahra Sadat Khosravi et al., in the article “Effect of Vitamin D3 Supplementation on Weight Loss, Glycemic Indices, and Lipid Profile in Obese and Overweight Women: A Clinical Trial Study”, concluded that supplementation with vitamin D3 reduced body weight, BMI, and abdominal circumference over 6 weeks [41]. Lida Lotfi-Dizaji et al., in their study with a 12-week duration [19], found significant results among groups, including a significant decrease in body weight and fat mass, as well as a significant increase in serum concentrations of 25(OH)D3 following supplementation with vitamin D3 compared with the group not supplemented with vitamin D3 [42].

Another study concluded that among healthy overweight and obese women, increasing the concentration of 25(OH)D3 through supplementation with vitamin D3 led to a reduction in body fat mass [43]. Low values are observed in all three combinations of heterozygous or homozygous polymorphisms on the *VDR* and *CYP2R1* genes and without polymorphisms on the *GC* gene, without mutations on the *VDR* gene, but with mutations on the *CYP2R1* and *GC* genes. Mutations on the *VDR* and *GC* genes, without mutations on the *CYP2R1* gene.

The analysis of the evolution of the risk for metabolic syndrome within the studied groups shows a significant improvement in metabolic health in the majority of the studied groups:

In the CG group, the number of subjects at low risk for metabolic syndrome decreased from two at the beginning of this study to one at the end, indicating an improvement in metabolic health in this group.

In the POW group, the reduction in the number of subjects at low risk for metabolic syndrome from fourteen to four at the end of this study shows a significant improvement in metabolic health in this group.

In the PO group, the majority of subjects remained at low risk for metabolic syndrome, with a slight decrease in their number from 16 to 13 at the end of this study.

In the PWS group, the number of subjects at low risk for metabolic syndrome decreased from fifteen to nine at the end of this study, indicating a significant improvement in metabolic health in this group. The interventions applied in these studies had a positive impact on the risk for metabolic syndrome, leading to a reduction or maintenance of the risk at lower levels in most groups. These findings suggest that lifestyle changes, including diet and supplementation with vitamin D3, can play an important role in the prevention and management of metabolic syndrome.

Reducing the risk of type 2 diabetes: vitamin D3 can improve insulin sensitivity, thus reducing the risk of insulin resistance and type 2 diabetes development, which are components of metabolic syndrome.

Lowering glucose and insulin levels: vitamin D3 supplementation can contribute to reducing blood glucose levels and improving pancreatic beta cell function, which secretes insulin. This can help maintain insulin levels within normal limits and prevent complications related to metabolic syndrome.

Decreasing cholesterol and triglyceride levels: vitamin D3 can influence lipid metabolism, helping to reduce total cholesterol levels and LDL (“bad”) cholesterol, as well as triglycerides. These effects can reduce the risk of cardiovascular diseases associated with metabolic syndrome.

Vitamin D3 supplementation may offer favorable impacts on metabolic syndrome, including enhanced insulin sensitivity, decreased levels of glucose and insulin, reduced blood lipids, and diminished inflammation. These effects contribute to the management and prevention of metabolic syndrome and associated complications.

By comparing the results obtained between the GC, POW, PO, and PWS groups, the specific impact of the genetic diet in the absence of vitamin D3 supplementation could be evaluated, and where supplementation was also present, providing important information regarding its effectiveness in managing certain metabolic or health conditions.

After evaluating the results obtained in the four study groups, we can conclude that vitamin D3 supplementation likely had a significant impact on metabolic health and associated risk factors for metabolic syndrome.

Initially, a notable decrease in total cholesterol was noted in the POW and PO groups after the administration of vitamin D3 supplements. This indicates that vitamin D3 may contribute positively to lowering the risk of cardiovascular diseases linked to elevated cholesterol levels. Additionally, a significant reduction in the HOMA-IR index was observed in the POW and PO groups, indicating an improvement in insulin sensitivity and better glucose management, crucial aspects in preventing type 2 diabetes mellitus and metabolic syndrome.

A significant decrease in abdominal circumference was noted in the POW (−6.81 cm), PO (−9.85 cm), and PWS (−7.68 cm) groups, on average, indicating a reduction in abdominal fat and, consequently, the risk of obesity and associated metabolic diseases, likely due to both the genetic diet and the administration of vitamin D3.

Overall, vitamin D3 supplementation had beneficial effects on reducing cholesterol, abdominal fat, and the HOMA-IR index in the studied groups, indicating promising potential in preventing and managing metabolic syndrome and associated diseases.

Vitamin D3 supplementation demonstrated positive effects in lowering cholesterol, abdominal fat, and the HOMA-IR index within the studied groups, suggesting promising prospects for preventing and managing metabolic syndrome and related conditions.

This positive response indicates the effectiveness of the applied treatment and the relevance of personalized interventions in managing health. It is important to note that adapting the dietary regimen to individual genetic characteristics and supplementing with vitamin D3, where applicable, contributed significantly to the improvement of health parameters in these groups. Personalized nutritional interventions and treatments tailored to the genetic characteristics of each individual may represent effective strategies for improving metabolic health and reducing the risk of associated conditions, such as metabolic syndrome and other metabolic diseases.

### Limitations of this Study

It may not be possible to generalize our findings to the Romanian population because of the sample’s shortcomings. At the same time, it should be noted that for the Romanian population, until now, no similar study has been carried out regarding the relationship between Vitamin D3 deficiency, metabolic syndrome, and *VDR*, *GC*, and *CYP2R1* gene polymorphisms, so this study represents a reference one that it will be able to continue for other age groups or other categories of patients.

## 5. Conclusions

The present study has confirmed a clear association between the presence of polymorphisms in the *VDR*, *CYP1R2*, and *GC* genes and low levels of vitamin D3. Achieving protective levels of 25(OH)D3, with values above 50 ng/mL, is similarly related to the presence of the studied polymorphisms. Changes in body weight, body mass index (BMI), and abdominal circumference have indicated that weight loss and reduction in BMI, as well as abdominal circumference reduction and better levels of total cholesterol, LDL cholesterol, and HOMA index, are all strongly associated with optimal concentrations of 25(OH)D3. This study’s results indicate that supplementation with vitamin D3, reaching an optimal level, can play a significant role in improving metabolic health and reducing the risk of associated conditions, such as metabolic syndrome.

Understanding the role of genes in regulating metabolism can be crucial in developing personalized approaches to the prevention and treatment of metabolic conditions, such as metabolic syndrome and diabetes mellitus. These findings may contribute to the development of more effective intervention strategies and the overall optimization of metabolic health management.

## Figures and Tables

**Figure 1 nutrients-16-01272-f001:**
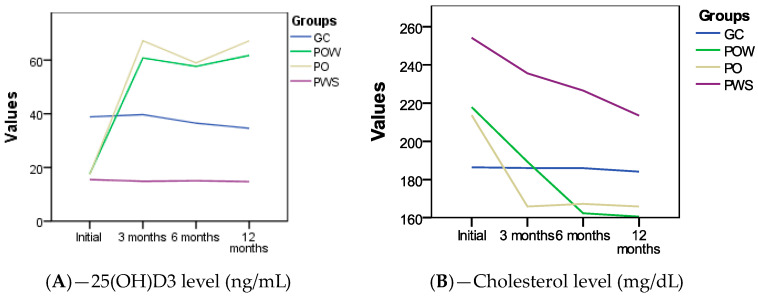
Evolution of 25(OH)D3 level (ng/mL) (**A**); total serum cholesterol (mg/dL) (**B**); HDL cholesterol level (mg/dL) (**C**); LDL cholesterol level (mg/dL) (**D**); triglyceride level (mg/dL) (**E**); HOMA index variation (**F**) in the studied groups; CG (control group) without *VDR*, *GC*, and *CYP2R1* gene polymorphisms, who followed a genetic diet; POW (overweight subjects group), with *VDR*, *GC*, and *CYP2R1* and gene polymorphisms, who followed a genetic diet and vitamin D3 supplementation; PO (obese subjects group) with *VDR*, *GC*, and *CYP2R1* gene polymorphisms, who followed a genetic diet and vitamin D3 supplementation; PWS (without vitamin D3 supplementation’s subjects group) with *VDR*, *GC*, and *CYP2R1* gene polymorphisms, who followed a genetic diet.

**Figure 2 nutrients-16-01272-f002:**
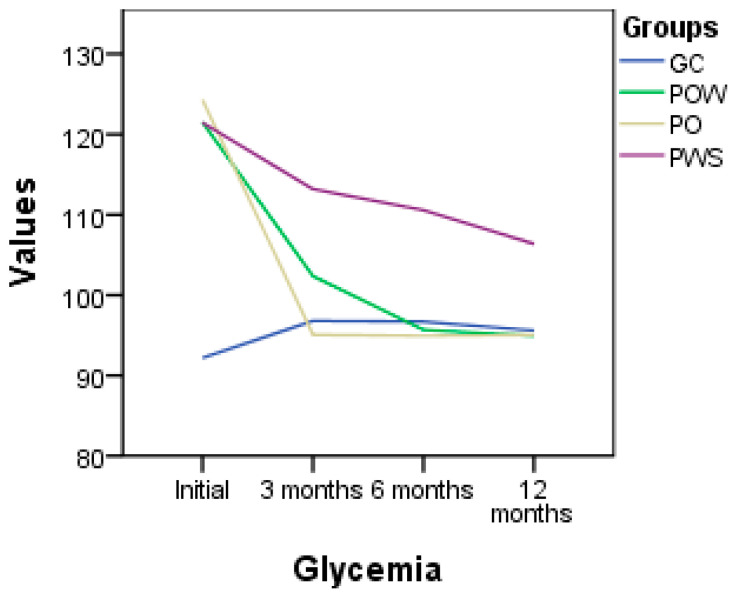
Glycemia level (mg/dL) variation in the studied groups; CG (control group) without *VDR*, *GC*, and *CYP2R1* gene polymorphisms, who followed a genetic diet; POW (overweight subjects group), with *VDR*, *GC*, and *CYP2R1* gene polymorphisms, who followed a genetic diet and vitamin D3 supplementation; PO (obese subjects group) with *VDR*, *GC*, and *CYP2R1* gene polymorphisms, who followed a genetic diet and vitamin D3 supplementation; PWS (without vitamin D3 supplementation’s subjects group) with *VDR*, *GC*, and *CYP2R1* gene polymorphisms, who followed a genetic diet.

**Figure 3 nutrients-16-01272-f003:**
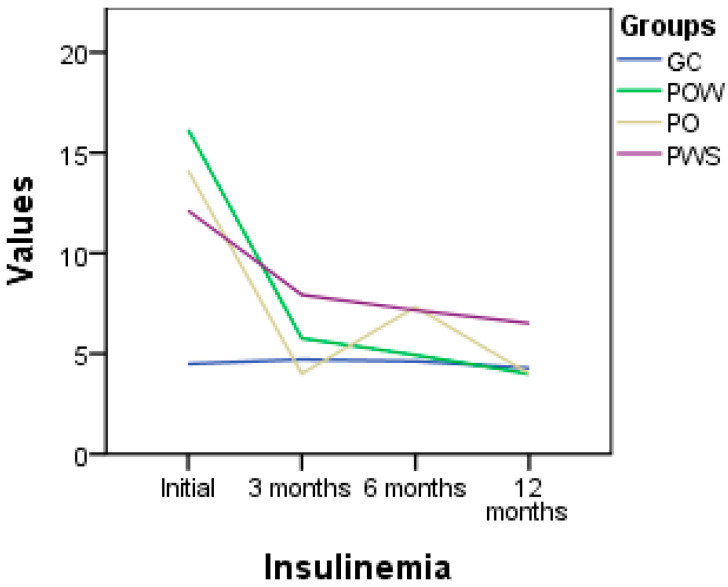
Insulin level (mUI/L) variation in the studied groups; CG (control group) without *VDR*, *GC*, and *CYP2R1* gene polymorphisms, who followed a genetic diet; POW (overweight subjects group), with *VDR*, *GC*, and *CYP2R1* gene polymorphisms, who followed a genetic diet and vitamin D3 supplementation; PO (obese subjects group) with *VDR*, *GC*, and *CYP2R1* gene polymorphisms, who followed a genetic diet and vitamin D3 supplementation; PWS (without vitamin D3 supplementation’s subjects group) with *VDR*, *GC*, and *CYP2R1* gene polymorphisms, who followed a genetic diet.

**Figure 4 nutrients-16-01272-f004:**
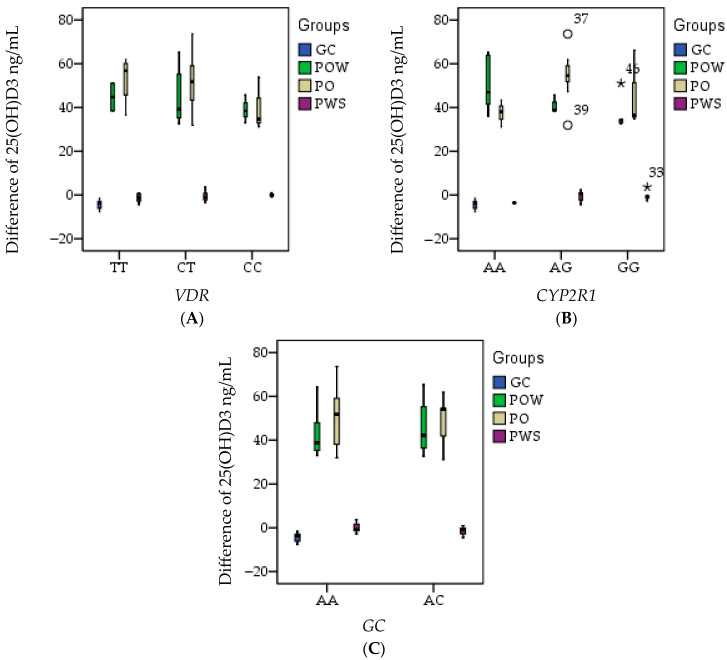
Graphical representation of 25(OH)D3 level (ng/mL) differences according to the presence of studied genes SNPs: *VDR* (**A**), *CYP2R1* (**B**), and *GC* (**C**) in the studied groups; CG (control group) without *VDR*, *GC*, and *CYP2R1* gene polymorphisms, who followed a genetic diet; POW (overweight subjects group), with *VDR*, *GC*, and *CYP2R1* gene polymorphisms, who followed a genetic diet and vitamin D3 supplementation; PO (obese subjects group) with *VDR*, *GC*, and *CYP2R1* gene polymorphisms, who followed a genetic diet and vitamin D3 supplementation; PWS (without vitamin D3 supplementation’s subjects group) with *VDR*, *GC*, and *CYP2R1* gene polymorphisms, who followed a genetic diet.

**Table 1 nutrients-16-01272-t001:** Vitamin D3 supplementation. The treatment was scheduled for the POW and PO groups over a 12-month period from entry into this study.

Baseline 25 OH Vitamin D3 Levels (ng/mL)	Vitamin D3	Total Initial Dose	Maintenance Dose Following Normalisation Between 50 and 70 ng/mL
10–19.99	200,000 IU week 1 100,000 IU week 4 100,000 IU week 8	400,000 IU	100,000 IU every 3 months
20–29.9	100,000 IU week 1 100,000 IU week 4 100,000 IU week 8	300,000 IU	
30–49.99	100,000 IU week 1 100,000 IU week 4 100,000 IU week 8	300,000 IU	

**Table 2 nutrients-16-01272-t002:** Demographic description of the studied groups.

Group	Average Age	Sex (Male/Female)	Origin (Urban/Rural)	*VDR*, *GC*, *CYP2R1* Gene Polymorphism (Yes/No)	25(OH)D3 Level (Average Value) Initial ng/mL	Initial BMI (Average Value)
CG	42.4	7/3 (70%/30%)	8/2 (80%/20%)	No	38.84	24.38
POW	37.37	14/2 (87.5%/12.5%)	10/6 (62.5%/37.5%)	Yes	17.58	27.14
PO	42.06	16/1 (93.75%/6.25%)	12/4 (75%/25%)	Yes	18.86	34.99
PWS	38.68	10/6 (62.5%/37.5%)	14/2 (87.5%/12.5%)	Yes	16.19	26.41

**Table 3 nutrients-16-01272-t003:** Genotype distributions and allele frequencies for the studied polymorphisms.

Gene	SNPs (Single Nucleotide Polymorphisms)	Genotype/Phenotype	Genotype/Phenotype	Genotype/Phenotype
*VDR*	rs731236	T/T—normal/higher Vit D3 level	T/C—lower Vit D3 level	C/C—lower Vit D3 level
*CYP2R1*	rs10741657	A/A—normal/higher Vit D3 level	A/G—normal/higher Vit D3 level	G/G—lower Vit D3 level
*GC*	rs2282679	A/A—normal/higher Vit D3 level	A/C—lower Vit D3 level	C/C—lower Vit D3 level

**Table 4 nutrients-16-01272-t004:** Frequency of the *VDR* gene polymorphism.

*VDR* Gene rs731236 SNP Genotypes	Frequency	%	*p*
T/T	22	37.9	0.001 **
C/T	27	46.6	
C/C	9	15.5	
Total	58	100.0	

*p* = statistically significance, ** = Correlation is significant at the 0.01 level (2-tailed).

**Table 5 nutrients-16-01272-t005:** Frequency of the *CYP2R1* gene polymorphism.

*CYP2R1* Gene rs10741657 SNP Genotypes	Frequency	%	*p*
A/A	21	36.2	0.042 *
A/G	21	36.2	
G/G	16	27.6	
Total	58	100.0	

*p* = statistically significance, * = Correlation is significant at the 0.05 level (2-tailed).

**Table 6 nutrients-16-01272-t006:** Frequency of the *GC* gene polymorphism.

*GC* Gene rs2282679 SNP Genotypes	Frequency	%	*p*
A/A	34	58.6	0.038 *
A/C	24	41.4	
Total	58	100.0	

*p* = statistically significance, * = Correlation is significant at the 0.05 level (2-tailed).

**Table 7 nutrients-16-01272-t007:** Evolution of BMI over the studied 12 months; CG (control group) without VDR, GC, and CYP2R1 gene polymorphisms, who followed a genetic diet; POW (overweight subjects group), with VDR, GC, and CYP2R1 gene polymorphisms, who followed a genetic diet and vitamin D3 supplementation; PO (obese subjects group) with VDR, GC, and CYP2R1 gene polymorphisms, who followed a genetic diet and vitamin D3 supplementation; PWS (without vitamin D3 supplementation’s subjects group) with VDR, GC, and CYP2R1 gene polymorphisms, who followed a genetic diet.

BMI Weight (kg)/Height (m^2^)	Group			
	GC	POW	PO	PWS
	Mean	Mean	Mean	Mean
BMI initial	22.83	27.14	34.99	29.58
BMI 3 months	22.92	24.48	30.81	27.69
BMI 6 months	22.89	24.22	30.66	26.91
BMI 12 months	22.90	24.18	28.92	26.41

**Table 8 nutrients-16-01272-t008:** Evolution of abdominal circumference in this study’s groups: CG (control group) without *VDR*, *GC*, and *CYP2R1* gene polymorphisms, who followed a genetic diet; POW (overweight subjects group), with *VDR*, *GC*, and *CYP2R1* gene polymorphisms, who followed a genetic diet and vitamin D3 supplementation; PO (obese subjects group) with *VDR*, *GC*, and *CYP2R1* gene polymorphisms, who followed a genetic diet and vitamin D3 supplementation; PWS (without vitamin D3 supplementation’s subjects group) with *VDR*, *GC*, and *CYP2R1* gene polymorphisms, who followed a genetic diet.

Group	GC	POW	PO	PWS
	Mean (cm)	Mean (cm)	Mean (cm)	Mean (cm)
Initial abdominal circumference	74.4	86.43	94.28	94.28
Abdominal circumference at 3 months	74.5	80.5	88.56	88.56
Abdominal circumference at 6 months	74.4	79.62	88.37	88.37
Abdominal circumference at 12 months	74.2	79.62	84.42	84.43

**Table 9 nutrients-16-01272-t009:** The Pearson correlation.

Pearson Correlation		VDR	CYP2R1	GC
25(OH)D3 level	r	0.296 *	0.014	0.124
	*p*	0.024	0.914	0.352
Cholesterol	r	−0.369 **	−0.484 **	−0.191
	*p*	0.004	0.000	0.152
HDL cholesterol	r	0.281 *	0.287 *	0.173
	*p*	0.033	0.029	0.193
LDL cholesterol	r	−0.249	−0.294 *	−0.152
	*p*	0.059	0.025	0.256
Triglycerides	r	−0.205	−0.342 **	−0.147
	*p*	0.123	0.009	0.272
Glycemia	r	−0.374 **	−0.146	−0.202
	*p*	0.004	0.273	0.128
	N	58		

r = Pearson coefficient, *p* = statistically significant, * = Correlation is significant at the 0.05 level (2-tailed), ** = Correlation is significant at the 0.01 level (2-tailed).

## Data Availability

The original contributions presented in the study are included in the article, further inquiries can be directed to the corresponding author.

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
