# Peer review of "Relationship between Vitamin D3 Deficiency, Metabolic Syndrome and VDR, GC, and CYP2R1 Gene Polymorphisms"

_nutrients, 2024, doi:10.3390/nu16091272_

Round 1

Reviewer 1 Report

Comments and Suggestions for Authors

The authors promote positive results in the prevention of abdominal obesity, highlighting the importance of personalized dietary approaches in promoting and regaining metabolic health. The authors presented the 12-month study, encompassing four distinct subject groups with 58 subjects. The study results show that all subjects in the groups with polymorphisms and supplemented with vitamin D3 reached an optimal level of vitamin D associated with high concentrations of 25(OH)D. Weight loss was the most significant in patients from the POW group (over 30 weight patients). Different groups adhered to a genetic diet regimen or additional vitamin D3 supplementation by a predefined protocol alongside the genetic dietary intervention.  

The study is well designed and suggests that for subjects with polymorphisms in vitamin D genes and components of metabolic syndrome, adopting a personalized genetic diet and vitamin D3 supplementation may be effective strategies for achieving optimal metabolic health and reducing the risk of metabolic syndrome. 

Additionally, it validated the importance of personalized approaches in managing health, considering individual genetic variations and their interactions with environmental factors.

Results suggest that interventions such as Vitamin D3 supplementation or specific study protocols may promote weight loss or improve weight management in individuals with higher initial BMIs.

Besides minor errors in the language spectrum, this paper has interesting results sufficiently elaborated. The results of such genetic diet and vitamin D3 supplementation represent effective interventions in reducing abdominal circumference and managing the risks associated with abdominal obesity and metabolic syndrome across different study groups.

Comments on the Quality of English Language

Minor errors in the language spectrum, text editing and misspelling.

Author Response

Response to Reviewer 1

We are very grateful for the effort and time you have devoted to this task. We, the authors of the present manuscript wish to thank you for thoughtful commentary you have provided to improve the quality of the paper.

We have extensively revised our manuscript. All changes in the text are highlighted.

Reviewer 2 Report

Comments and Suggestions for Authors

This study investigates the possible relationship between vitamin D deficiency, MS and VDR, GC, CYP2R1 genes polymorphism, and genes whose encoded 22 proteins are responsible for vitamin D metabolism and transport.

Some interesting conclusions could be drawn by the study:

Genetic polymorphisms play a significant role in the evolution of D3 levels after supplementation

The genetic diet of group PWS could not help patients without D3 supplementation elevate D3 levels.

Genetic diet and D3 supplementation helped participants obese and overweight improve HDL levels, lower LDL and total cholesterol levels, their glucose and insulin levels and body weight.

The authors conclude that individuals with polymorphisms and some components of metabolic syndrome  should follow a genetic diet as well as D3 supplementation to achieve optimal health and reduce the risk for developing metabolic syndrome.

Some points though should be addressed to y the authors:

1)      The study included 58 volunteers divided in four groups depending on their gene polymorphisms which is a rater small sample for such a study taking in account the diversity of the population.

2)      Authors should also define the “genetic diet” followed by the participants (l167-174).

3)      The authors do not define what was the total number of subjects tested so as to determine the occurrence of polymorphisms in the total number (the 10 participants of the CG group were the only ones found, making the sample with polymorphisms almost 1/6 of the total population or were just 10 included from a larger total sample)?

Comments on the Quality of English Language

Minor editing or grammatical and syntax errors should be corrected. Overall quality of english language is adequate for publication.

Author Response

Response to Reviewer 2

We are very grateful for the effort and time you have devoted to this task. We, the authors of the present manuscript wish to thank you for thoughtful commentary you have provided to improve the quality of the paper.

We have extensively revised our manuscript according to the recommendations. All changes in the text are highlighted. Please, see the point-by-point answers to your comments below.

  • The study included 58 volunteers divided in four groups depending on their gene polymorphisms which is a rater small sample for such a study taking in account the diversity of the population.

Thank you for your comment, the 58 volunteers were selected from the nutrition cabinet. SC CSBNDiet Precision NUTRITION SRL, with the aim of investigating the evolution of patients to genetic polymorphisms in the context of a genetic diet and/or supplementation with vitamin D. We acknowledge that although our sample size is not large, it was chosen taking into account the specific demographics of the population and the focused nature of our study objectives. This approach allowed us to perform a detailed analysis within the defined scope of our research, providing valuable insights into the interaction between genetic polymorphisms, dietary interventions and vitamin D status.

  • Authors should also define the “genetic diet” followed by the participants (l167-174).

Thank you for your comment, we have completed in the text the definition of  the genetic diet

3)   The authors do not define what was the total number of subjects tested so as to determine the occurrence of polymorphisms in the total number (the 10 participants of the CG group were the only ones found, making the sample with polymorphisms almost 1/6 of the total population or were just 10 included from a larger total sample)?

Thank you for your comment, the study was conducted among patients who presented at the nutrition cabinet. SC CSBNDiet Precision NUTRITION SRL, with polymorphisms in the 3 genes VDR, CYP2R1, and GC and without polymorphisms were included in the study.

The CG group represents the control group, it was created from patients without polymorphisms in the VDR, CYP2R1 and GC genes, in order to assess the results as accurately as possible. The results could be statistically interpreted and highlight the importance of investigating polymorphisms for a deeper understanding of genetic predispositions associated with nutritional response.

As a limitation of the study we specified that it may not be possible to generalize our findings to the Romanian population because of the sample's shortcomings. At the same time, it should be noted that for the Romanian population, until now no similar study has been carried out regarding the relationship between Vitamin D deficiency, metabolic syndrome and VDR, GC, CYP2R1 gene polymorphism, so this study represents a reference one that it will be able to continue for other age groups or other categories of patients. We intend to continue studies in the general population.

Round 2

Reviewer 2 Report

Comments and Suggestions for Authors

Since all points and suggestions have been answered and have been incorporated to the revised document I have no further suggestions to make.

Author Response

We, the authors of the present manuscript, are deeply grateful and thank you for your valuable comments and by which you have helped us to improve the quality of the paper.